# Omicron Genetic and Clinical Peculiarities That May Overturn SARS-CoV-2 Pandemic: A Literature Review

**DOI:** 10.3390/ijms23041987

**Published:** 2022-02-11

**Authors:** Giorgio Tiecco, Samuele Storti, Melania Degli Antoni, Emanuele Focà, Francesco Castelli, Eugenia Quiros-Roldan

**Affiliations:** Unit of Infectious and Tropical Diseases, Department of Clinical and Experimental Sciences, ASST Spedali Civili di Brescia and University of Brescia, 25123 Brescia, Italy; g.tiecco@unibs.it (G.T.); s.storti@unibs.it (S.S.); m.degliantoni@unibs.it (M.D.A.); emanuele.foca@unibs.it (E.F.); francesco.castelli@unibs.it (F.C.)

**Keywords:** Omicron, B.1.1.529, variants of concern, SARS-CoV-2, COVID-19

## Abstract

The Coronavirus disease 2019 (COVID-19) pandemic poses a great threat to global public health. The original wild-type strain of severe acute respiratory syndrome coronavirus 2 (SARS-CoV-2) has genetically evolved, and several variants of concern (VOC) have emerged. On 26 November 2021, a new variant named Omicron (B.1.1.529) was designated as the fifth VOC, revealing that SARS-CoV-2 has the potential to go beyond the available therapies. The high number of mutations harboured on the spike protein make Omicron highly transmissible, less responsive to several of the currently used drugs, as well as potentially able to escape immune protection elicited by both vaccines and previous infection. We reviewed the latest publication and the most recent available literature on the Omicron variant, enlightening both reasons for concern and high hopes for new therapeutic strategies.

## 1. Introduction

The Coronavirus disease 2019 (COVID-19) pandemic poses a great threat to global public health—more than 260 million confirmed cases have been reported, resulting in over 5 million deaths [1]. The original wild-type strain of severe acute respiratory syndrome coronavirus 2 (SARS-CoV-2), identified at the end of 2019 in Wuhan, has since genetically evolved, and several variants have emerged. Until late 2021, four variants of concern (VOC) of SARS-CoV-2 had been described, including Alpha (B.1.1.7), Beta (B.1.351), Gamma (P.1), and Delta (B.1.617.2). On 26 November 2021, a new variant named Omicron (B.1.1.529) was designated the fifth VOC [2]. This new VOC harbours a significant number of mutations on the spike protein (S protein) and appears to be highly transmissible as well as potentially able to escape immune protection elicited by both vaccines and previous infection [2,3]. Although recent data suggest that Omicron has a less severe clinical presentation [4,5,6], it is still too early to conclude on the clinical impact of this variant. Furthermore, the increased transmissibility and the consequent exponential growth of Omicron cases have currently made the Omicron variant the protagonist of the ongoing new surge. Hereby, we review the latest publication and the most recent available literature on the Omicron variant.

## 2. Virology and Pathogenesis

Phylogenetic studies reveal that the Omicron variant has likely diverged early from other SARS-CoV-2 strains [7] (Figure 1). It is even speculated that the Omicron variant might have been gestated in immunocompromised individuals (e.g., HIV patients coinfected by SARS-CoV-2) for a while [8]. However, according to recent studies based on nonsynonymous mutations analysis in Omicron open reading frame (ORF), the molecular spectrum of preoutbreak mutations is inconsistent with the rapid accumulation of mutations in humans. Moreover, the B.1.1 variants and human coronavirus hCoV-229E show the highest sequence similarities [9]. These results strongly support a trajectory in which the progenitor of Omicron experienced a reverse zoonotic event from humans to mice during the pandemic accumulating mutations [9].

The Omicron variant is not structurally different from other already identified SARS-CoV-2 VOCs. The S protein remains a trimer, and each monomer is composed of two subunits (S1 and S2). The receptor-binding domain (RBD), which interacts with the ACE2 receptor, is located in the S1 subunit (Figure 2) [10].

While these aspects are shared among VOCs, Omicron genomic features are highly divergent. Recently, whole-genome examination and mutational analysis have found that the Omicron variant might be classified into two different lineages (BA.1 and BA.2). Six genome sequences of both BA.1 and BA.2 were analysed and compared to the original Wuhan strain, and 32 common mutations were found in both lineages, whilst 19 mutations were lineage-specific and considered “signature mutations” [11].

More specifically, 21 common mutations and 13 signature mutations have been found in the BA.1 lineage S-glycoprotein (34 mutations overall), whereas 7 signature mutations are described in the BA.2 lineage S-glycoprotein (28 mutations overall) [11].

Considering coding and noncoding regions, Omicron carries at least 60 mutations in total compared with the original Wuhan strain. Many of those mutations are shared with the previously described VOCs, whereas others are uniquely found in Omicron (Figure 3). Similarly, there are 6 deletions (in positions 69, 70, 143, 144, 145, and 211) in the Omicron variant, of which only the one in position 211 does not appear in the other VOCs [12]. Crucial implications in infectiousness and immune escape regard at least 36 mutations in the S protein; over 30 amino acid substitutions, 3 deletions and 1 insertion are recorded. Notably, 15 of the 30 amino acid substitutions are in the receptor-binding domain (RBD) [13]. Lastly, the Omicron variant shares the mutations at the furin cleavage region with the other VOC. During the S protein cleavage process, the mutations in positions 547, 655, 679 and 681 allow the formation of two subunits which boost the transmissibility of the virus [12].

The Omicron variant shows a three to four-fold increase in the number of mutations expressed on the spike protein compared with the other 4 VOCs [2]. Furthermore, seven common mutations (G142D, K417N, T478K, N501Y, D614G, H655Y, and P681H) and three signature BA.1 lineage mutations (ΔHV69del, T95I, and ΔYY144del) overlap Alpha, Beta, Gamma, and Delta VOC. These overlapping mutations have been previously associated with increased transmissibility, more efficient viral binding, as well as immune evasion. The D614G mutation correlates with a higher upper respiratory tract viral load and a younger age of the patients affected, and it has been found in all five VOCs already identified. Furthermore, the Omicron variant shares N501Y, which is believed to increase the binding affinity between the viral spike protein and the angiotensin-converting enzyme 2 (ACE2) receptor [2].

Regarding the structural proteins, three substitutions (D3G, Q19E, and A63T) involve the membrane, one substitution (T9I), the envelope and three substitutions, and a three-residue deletion of the nucleocapsid proteins [2].

Lastly, another crucial point regards the Omicron entry pathway, which might have implications on the clinical manifestations and disease severity. While the Delta variant replicated well in Calu-3 cells, which has robust TMPRSS2 (transmembrane serine protease 2) expression, the Omicron variant replicated poorly in this cell line, showing a weaker cell–cell fusion activity [14]. As a matter of fact, Camostat, which inhibits the TMPRSS2 pathway alone, significantly reduced only the Delta variant entry pathway. This observation supports the point that the Omicron variant infection is not enhanced by TMPRSS2 but is largely mediated by the endocytic pathway instead [14]. The inefficiency in using TMPRSS2 might explain the dramatically attenuated replication rate in Calu3 and Caco2 cells with a less severe lung pathology [15].

### 2.1. Transmissibility

Data from countries with an early spread of Omicron suggest that Omicron is more transmissible than the Delta variant [16]. Genetic aspects might explain this concerning the Omicron feature:N501Y mutation increases binding to the ACE2 receptor, especially when associated with Q498R [17] or the H69/V70 deletion [2]. The unique dual mutation N501Y and Q498R found in the Omicron variant, combined with E484K and S477N mutations, might increase the affinity to ACE2 receptor by up to 1000-fold and up to the level of low pM in KD value [18]. This is also explained by studies conducted using computational modelling and simulations: due to N501Y mutation, a higher number of hydrogen bonds is formed (6.5 ± 2.2) between RBD and ACE2 receptors [19].The positions of several mutations (H655Y, N679K), in relation to the furin cleavage site, are supposed to enhance spike cleavage (S1/S2 junction) and aid transmission. Furthermore, the P681H mutation, already found in the Alpha and similarly (P681R) in the Delta VOC, might increase the transmission rate through the same mechanism [18].Lastly, the presence of R230K and G204R mutations in the nucleocapsid was linked to an increased viral load and might be a major modulator of host–virus interactions of not fully understood significance [20].

However, according to the deep mutational scanning test, other substitutions in the Omicron spike protein (K417N, G446S, E484A, Q493R, G496S, and Y505H) are expected to decrease the binding to ACE2 receptor [18,19,20,21]. As a result, Omicron’s RBD only has a 2.4-fold increased binding affinity to human ACE2 [18]. It goes without saying that RBD is just one of the many aspects that might impact virus transmissibility, and other mutations may still occur in the VOC.

For example, the same mutational pattern that reduces affinity to the ACE2 receptor (G446S, Q493R, and G496S) may also create steric interference for the binding of antibodies to the RBD, whereas E484A, Y145del, and Y505H may result in the complete loss of interactions between an antibody and RBD, raising concern for immune escape and reinfections [22].

### 2.2. Immune Evasion

In contrast to the Beta or Delta VOC, population-wide epidemiological evidence and several studies demonstrated that the Omicron variant is associated with a significant ability to escape humoral immunity [23]. This might explain why even three doses of mRNA vaccines may not be sufficient to prevent infection and symptomatic disease with this VOC [24].

In comparison with the Delta VOC, the increased number of mutations located in the Omicron N-terminal domain (NTD) and RDB dramatically modify exposed epitopes, making them difficult to be recognised by the NTD-targeting neutralising antibodies, allowing reinfection, and reducing the efficacy of currently used vaccine (Table 1) [12]. Moreover, mutations as S371L, N440K, G446S, and Q493R in the S protein confer a greater antibody resistance to Omicron related to previously known VOCs [25,26].

Furthermore, in the NTD, a unique combination of mutations (Δ211/L212I, ins214EPE) seems to disrupt an HLA class II epitope impairing dendritic cell priming [27]. However, in vitro assay system showed that, although the neutralisation sensitivity of convalescent sera decreased, the average mean neutralisation titre defined by the ED50 (50% effective dilution) against Omicron is still higher than the baseline. This suggests that there is still some protection effect [3]. It is crucial to underline that, although in this assay, the real virus has not been employed, vaccine literature on previously known VOC has established that the in vitro neutralisation assays are good predictors of in vivo and real-life immunization [28].

Humoral immunity escape may also be linked to a higher risk of reinfection (defined as a repeat positive test at least 90 days after an earlier positive test) in individuals previously infected with a different strain [23]. This is also supported by preliminary findings from several laboratories in which sera from previously infected, unvaccinated individuals, did not neutralise the Omicron variant [29]. Whether subneutralising antibodies (from the previous infection with other VOC or vaccines) could facilitate the Omicron entry in the cells by their Fc receptor through the known antibody-dependent enhancement (ADE) mechanism remains to be investigated.

Notwithstanding, there are conflicting results about Omicron’s ability to escape the immune system. Most T cell epitopes seem to be preserved in Omicron, suggesting that CD8+ T cell responses and vaccines may remain protective against this VOC [18]. Moreover, although no study is already available on the Omicron variant, pre-existing nonspike cross-reactive memory T cells seem to protect SARS-CoV-2-naïve contacts from infection [30]. Finally, in contrast with previously known variants, it has been recently proposed that Omicron could be less effective in antagonising the interferon produced by the host cells in response to the infection. Authors suggest that this might explain why Omicron causes a milder disease compared with other VOCs [31].

### 2.3. Replication Advantage

The Omicron relative instantaneous reproduction number (RRI), defined as the ratio of the effective reproduction number of the target variant (Omicron) to that of the baseline variant (Delta), was found to be 3.19 (95% CI 2.82–3.61) times greater than that of Delta under the same epidemiological conditions [16]. Although these estimates are consistent with others found in Gauteng, South Africa [32], data were limited by being wholly based on the nucleotide sequences submitted to the GISAID database from Denmark [33].

Genetic modifications associated with this replication advantage regards both the S protein (Δ69-70, P681H, and H655Y) and the nucleocapsid protein (R203K and G204R) of the Omicron variant [34]. Although the effects of these mutations were individually demonstrated in previously known VOCs, the combination of all these variations in the same genome raises concerns about a supposed additive replication advantage of the Omicron variant.

### 2.4. Invasiveness

Early reports of patients infected with Omicron suggests that it produces a less severe disease when compared with previously known VOCs. However, it is difficult to establish whether Omicron causes milder disease or not. Omicron replicates faster in the upper airways rather than in the lung, and, consequently, reduced levels of the virus have been found in human and animal models’ lung tissue [35,36].

## 3. Epidemiology

Very recently, there was an alarming increase in COVID-19 cases in South Africa: in November 2021, the mean number of COVID-19 cases per day increased from 280 to 800 [2]. On 24 November 2021, the identification of a new SARS-CoV-2 variant, B.1.1.529, was reported by the southern African authorities to the World Health Organization (WHO) [37]. The first B.1.1.529 case was detected in specimens collected in Botswana on 11 November 2021 [38]. This VOC was soon promptly identified in multiple other nations, spreading to neighbouring countries such as Botswana, Namibia, Zimbabwe, Swaziland, and Mozambique [39]. Although several countries had arranged traveller restrictions for passengers from endemic areas, as of mid-December 2021, the Omicron variant accounted for the majority of new infections in the United States [37], and several studies showed that an international travel history within 14 days of symptom onset is not a must for Omicron community spreading [40]. However, according to analysis carried out by the National Wastewater Surveillance System, the detection of Omicron-associated mutations in community wastewater dates to at least a week before the first case was identified in the U.S. This suggests the possibility that variant tracking data from wastewater can be used as a complement to clinical testing for the early detection of emerging variants [41]. The second Omicron case in Europe was reported In North Italy in a patient travelling from Mozambique [42].

At the time of writing, the overall case notification for the EU/EEA was 2621/100,000 and had been increasing for four weeks, as had the ICU admissions. However, the COVID-19 14-day death rate has stabilised over the last eight weeks. The Omicron variant has become the dominant variant, although both Delta and Omicron are currently cocirculating [43]. Overall, VOCs epidemiological situation is currently being monitored by a genomic surveillance system in both Europe and the United States, which constitutes a key component of the public health efforts throughout the current pandemic [44].

## 4. Clinical Manifestations

The most common symptoms reported in confirmed cases of the Omicron variant infection are nasal congestion (73%), cough (65%), headaches (54%), sore throat (48%), chills (34%), and fever (32%). Only 10% reported shortness of breath, and most asymptomatic patients were vaccinated [45].

In South Africa, several studies accounted that the rate of in-hospital death, oxygen and mechanical ventilation requiring intensive care unit admission, and length of stay was lower among patients hospitalised with COVID-19 during the Omicron surge [46,47]. All these preliminary analyses reported that younger patients (median age, 36 years, *p* < 0.001), with fewer comorbidities, were mostly involved in this fourth wave. Furthermore, a higher proportion of unvaccinated patients in hospital admission (66.4%) was also described [48]. Similarly, more emerging reports suggest that Omicron is associated with milder illness: for example, in England, the risk of in-hospital admission with Omicron was approximately one-third of that associated with Delta, adjusted for age, sex, vaccination status, and prior infection [4]. These results are confirmed by several animal models that show lower viral levels in lung tissue and milder clinical features [35,49].

However, it is imperative to point out that not only these data are preliminary and potentially uncertain, but also that even if the individual risk for severe disease with Omicron is lower than with prior variants, the higher transmissibility can still result in an unbearable burden for the health care system [50]. The reported case of transmission of the Omicron variant between two patients staying in different rooms in a designated quarantine hotel in Hong Kong, despite the strict quarantine precautions, is iconic [51]. This case has brought scientists to demonstrate, through a smoke test, that even a door left slightly open might cause an inward airflow that could lead to infection [52].

## 5. Diagnosis

The number of mutations involving epitopes of the Omicron variant have made several single-target molecular tests ineffective, raising the false negative rate results in patients infected by this VOC [53]. While Delta VOC is defined as S-gene positive, the Omicron variant carries the deletion at H69 and V70 in the spike gene, as does the Alpha VOC. This deletion in the spike protein results in an S-gene dropout and the inability of some SARS-CoV-2 molecular tests to detect the S-gene [54]. However, using multiple target molecular tests, the overall sensitivity should not be impacted [53]. Moreover, specimens that yield an S-gene target failure, when tested with these kits, might be used as a rapid proxy for the frequency of Omicron cases. New generation sequencing verification is always advised [55]: testing positive for SARS-CoV-2 with one of these tests does not mean an individual is infected with the omicron variant and, not every sample obtained from patients affected by the omicron variant displays a mutation that leads to a gene dropout [53].

Early data based on patient samples containing live viruses suggest that antigen tests, relying mainly on nucleocapsid proteins, can also detect the Omicron variant proteins but may have a reduced sensitivity [53]. Although these are encouraging preliminary results, it is crucial to underscore that antigen tests are generally less sensitive and less likely to pick up very early infections compared with molecular tests. Following the FDA’s long-standing rapid test recommendations, a negative antigenic test in a symptomatic person or with a high likelihood of infection due to exposure to a confirmed COVID-19 case requires, with a high degree of recommendation, a follow-up molecular testing [55].

## 6. Treatment

Currently, several therapeutic options approved under the FDA-issued Emergency Use Authorization (EUA) are available in COVID-19 management, including antiviral drugs (e.g., molnupiravir, nirmatrelvir, remdesivir), anti-SARS-CoV-2 monoclonal antibodies (e.g., bamlanivimab-etesevimab, casirivimab-imdevimab, and sotrovimab), anti-inflammatory drugs (e.g., dexamethasone), and immunomodulators agents (e.g., baricitinib, tocilizumab). For most of them, little data are available regarding their efficacy against the Omicron variant.

### 6.1. Monoclonal Antibodies

Every monoclonal antibody (mAb) against SARS-CoV-2 used so far binds to the virus’ S protein, preventing infection of human cells and reducing the risk of severe COVID-19 up to 85%. The shift in epitopes showed in the Omicron’s S protein (particularly within the RDB domain) raised concern about the effectiveness of some mAbs currently available in terms of susceptibility and potency [56].

Bamlanivimab–etesevimab (LY-CoV016-LY-CoV555) is a cocktail of 2 antibodies. Both bind to an overlapping epitope in the RBD of the SARS-CoV-2 S protein, targeting both the open and closed conformation increasing the neutralisation rate and blocking the attachment to the human ACE2 receptor. Similarly, casirivimab-imdevimab (REGN-CoV2), authorised as an intravenous or subcutaneous injection, reduces the risk of severe COVID-19 by binding distinct epitopes of the S protein [10]. The neutralisation potency of both cocktails is greatly reduced in the Omicron variant [56,57]. In vitro analysis using Vero-TMPRSS2 and Vero-hACE2-TMPRSS2 cells demonstrated a complete loss of inhibitory activity against this VOC [58]. As the Delta VOC still represents a major concern, bamlanivimab-etesevimab and casirivimab-imdevimab are still considered treatment options, but since Omicron is resistant to both, SARS-CoV-2 genotyping might be recommended before initiating mAbs treatment [59]. Even the promising regdanvimab (CT-P59) seems likely to be ineffective against the Omicron variant. Through the measurement of the median fluorescence intensity of the signal, regdanvimab displayed a strong reduction in its binding to Omicron infected cells when compared with Delta [60].

Sotrovimab (GSK4182136 or S309) binds to a cryptic RBD epitope shared between SARS-CoV and SARS-CoV-2 where, apparently, fewer mutations have occurred [56]. Since this mAb does not overlap with ACE2 on the RBD binding interface, its neutralising activity does not directly interfere with ACE2 binding [61]. Sotrovimab seems to be active against the Omicron variant, but its neutralising potency results dropped 3-fold [18]. High hopes come from a recently found RBD-specific antibody called bebtelovimab (LY-CoV1404) which potentially neutralises both the authentic SARS-CoV-2 virus (B.1.1.7, B.1.351 and B.1.617.2) and VOCs (B.1.1.7, B.1.351, B.1.617.2, B.1.427/B.1.429, P.1, and B.1.526), including Omicron (B.1.1.529). In this preliminary analysis, the binding and neutralising activity seems to be unaffected by the most common mutations presented by already known VOCs, making bebtelovimab a viable therapeutic agent for the treatment of the Omicron variant [62]. Lastly, from preliminary analysis, another newly synthesised mAb known as DXP-604 seems to be insensitive to the K417N single site change in the Omicron variant RBD. However, when K417N is combined with other mutations (S477N, Q493R, G496S, Q498R, N501Y, and Y505H), the DXP-604’s binding affinity against Omicron RBD is largely decreased (nearly 30-fold reduction compared with the wild-type strain) [57]. A panel of over 30 mAbs against Omicron are currently being tested with the primary goal of identifying those that retain a neutralising activity against this VOC and perhaps also modelling a new generation of mAbs [63].

### 6.2. Antivirals

Regulatory authorities are continuously evaluating already approved drugs, and hundreds of medicines with newly identified targets have been proposed [64]. The effect of approved drugs and those under investigation against the Omicron variant remains to be further investigated.

Remdesivir was the first FDA-approved drug in COVID-19 treatment. It blocks viral replication by acting as a nucleoside analogue inhibiting the RNA dependent RNA polymerase (RdRp) of Coronavirdae. Although no definitive study has been carried out on the Omicron variant, typical mutations in the RdRp associated with remdesivir resistance (e.g., V557L, V473F, N491S, F480L/S/C, P323L, or E802D) have not been described in this VOC yet [65]. However, new evidence on remdesivir intracellular signalling pathways is currently emerging as the possible implications on herpesviruses (HHV8 and EBV) reactivation [66]. Notwithstanding, a very low level of overall remdesivir resistance is described [67].

Molnupiravir is an isopropylester prodrug of the nucleoside analogue β-d-N4-hydroxycytidine (NHC), which increases the frequency of viral RNA mutations impairing SARS-CoV-2 replication. NHC-triphosphate, the active form of molnupiravir, is incorporated in viral RNA instead of cytidine triphosphate or uridine triphosphate. Moreover, this modified RNA, when used as a template, directs the incorporation of either G or A, leading to mutated RNA products [68]. Concerns regarding the possible mutational effect on uninfected human cells derive from in vitro studies on mammalian cells, which need further investigations [69]. Molnupiravir is orally available and, when administered early in non-hospitalised, unvaccinated adults with mild to moderate COVID-19, it reduces the overall hospitalisation and death risk by approximately 31% [70]. No data on the Omicron variant are yet available, but molnupiravir is expected to retain activity against all SARS-CoV-2 VOCs [64].

The antiviral drug nirmatrelvir (PF-07321332) will soon be available: it inhibits the SARS-CoV-2 protease (essential for viral replication) and is coadministered with ritonavir to slow its metabolism, allowing for longer persistence and higher drug concentrations [71,72]. Preliminary data on patients at high risk for severe COVID-19 show reduced hospital admissions and deaths by 80–90% [73]. The main problem is linked to ritonavir, a potent cytochrome and P-glycoprotein inhibitor, raising concerns about drug–drug interactions. The combination is expected to retain activity against all the SARS-CoV-2 variants, including the new Omicron [64].

The few data available about antiviral drugs suggest that Omicron remains sensitive to the approved antiviral drugs and several drug candidates with new mechanisms of action, such as nafamostat, camostat, or aprotinin [31].

Although in the history of infectious diseases, we have always benefited from the use of anti-infective combination therapies to treat several microorganisms (i.e., *M. tuberculosis*, HIV, HCV, or *Plasmodium* spp.), no data about combination therapies employing these drugs are currently available.

## 7. Impact of Vaccines on Omicron Infection

Vaccine-induced immunity targets the spike protein, which is, as said before, particularly mutated in the Omicron variant. Data from other VOCs with similar spike protein substitutions have shown a significant reduction in neutralising the activity of sera from vaccinated or previously infected individuals [13]. This might explain the rising cases in several countries despite an almost complete vaccinal coverage in the population [16]. Several reports demonstrate that the neutralising activity of sera from vaccinated individuals who received only a primary vaccine series is reduced and sometimes undetectable against Omicron compared with the wild-type virus or the Delta variant [74,75].

In South Africa, two doses of BNT162b2 were associated with a 33% effectiveness against SARS-CoV-2 infection and 70% effectiveness against COVID-19 hospitalisation during the Omicron surge [76]. Moreover, neutralising antibodies titre against Omicron variant were significantly lower when compared to titres against other VOCs [29]. In line with these findings, a recent analysis suggested that primary vaccination was insufficient to effectively neutralise Omicron spike proteins, allowing this VOC an efficient escape from humoral immunity [77].

Emerging data indicate that Omicron infection occurs even in individuals who had received a full primary vaccination series and a booster dose with mRNA vaccines [24]. However, the additional booster mRNA vaccine dose generates a highly effective antibody response, although 4–6-fold lower than against the wild-type strain of SARS-CoV-2 [77].

In comparison with the Delta VOC, a recent study suggests that unvaccinated or single-dose vaccinated people were more likely to be infected by Delta rather than Omicron, whereas people who had received a full primary vaccination cycle or had received the booster dose resulted in a 2–3 times higher odds of having Omicron [57,78].

Although these analyses state that even three doses of mRNA vaccines might not be sufficient to prevent infection, most patients experienced milder clinical manifestations than unvaccinated or not fully vaccinated people [24]. Effectiveness against hospitalisation remained similar when compared to Delta, especially among those who received a booster dose [76].

Despite the reduced neutralising activity of vaccinated individual sera against the Omicron variant, persistent protection against severe disease might be explained by the fact that vaccine- or infection-induced cellular immunity appears robust against the Omicron variant [79,80]. It is crucial to emphasize that vaccine effectiveness against symptomatic Omicron variant infection, and hospitalisation decreases over time, as stated by numerous studies [4,75]. A steeper decline in neutralising titres was observed among males over 65 years old and immunocompromised individuals [81]. The lower neutralising activity of vaccinated or infected by natural COVID infection, together with the gradual decrease of immunization within six months, might worsen the pandemic situation [3].

Cellular immunity has a central role in COVID-19 protection, and robust T cell responses have been associated with less severe disease [82]. A preserved T cell activity has been shown in sera of both previously infected and vaccinated, providing an extensive immune coverage against the Omicron variant [83,84]. Furthermore, recent data suggest that a pre-existing nonspike memory T cell could exist, allowing the development of second-generation vaccines based on nonspike antigens [30]. In contrast, extremely vulnerable patients, such as those undergoing haemodialysis, seem to have a reduced cellular response to SARS-CoV-2 vaccines, and a three doses course could be insufficient against Omicron [85].

Lastly, it is known that widespread vaccination reduces the overall transmission risk since vaccinated individuals are less likely to develop an infection. However, during Delta spreading, the difference between secondary attack rates from a vaccinated (25%) versus unvaccinated (38%) index case was not statistically significant [86]. Moreover, levels of upper respiratory tract SARS-CoV-2 RNA in vaccinated individuals were similar to those found in unvaccinated individuals with the Delta variant infection [87,88]. These concerns also apply to the Omicron variant, but, up to date, there is no available analysis about the correlation between the risk of transmission and vaccination coverage during the Omicron surge.

## 8. Conclusions

The Omicron variant has revealed that SARS-CoV-2 has the potential to go beyond the available therapies. Thus, to avoid an extenuating chase toward the virus changes, future COVID-19 therapy should ideally include several features such as high effectiveness in reducing viral load and minimising viral spreading, broad-spectrum protection against all VOCs, and a high resistance barrier [18]. Appropriate treatment and management strategies should necessarily be complemented with new rapid tools for predicting the affinity and interactions between ACE2 and RBD, gaining clues about the transmissibility and virulence of new variants to develop new diagnostic kits and vaccines [89].

However, is Omicron the last variant of concern? The new wave of Omicron, along with vaccination programs, is reducing the speed of the virus spreading in many parts of the world. However, it cannot be excluded that more variants may be seen in the future, and we are not yet able to predict whether they will escape this immunity. There are many more letters in the Greek alphabet, and variants with the ability to evade immunity and spread successfully are likely to emerge [90]. Perhaps, the next variant may already be circulating. In fact, while the world is grappling with the Omicron spreading, a new variant of COVID-19 known as IHU (B.1.640.2) has been recently identified in France [91].

Omicron’s surge and the presumable future spread of other VOCs reflect the wealthy nations’ failure to implement global strategies to reduce and avoid vaccination hesitancy and disparity [92]. By delaying this kind of action, more time to evolve is given to the virus, prolonging the pandemic and its social and economic consequences [93]. The WHO slogan states, “none of us is safe until all of us are safe” [94]. Hence, the introduction of public health and social measures in a targeted and consistent manner is necessary to stop the emergence of novel SARS-CoV-2 variants and ultimately eradicate this pandemic [93].

## Figures and Tables

**Figure 1 ijms-23-01987-f001:**
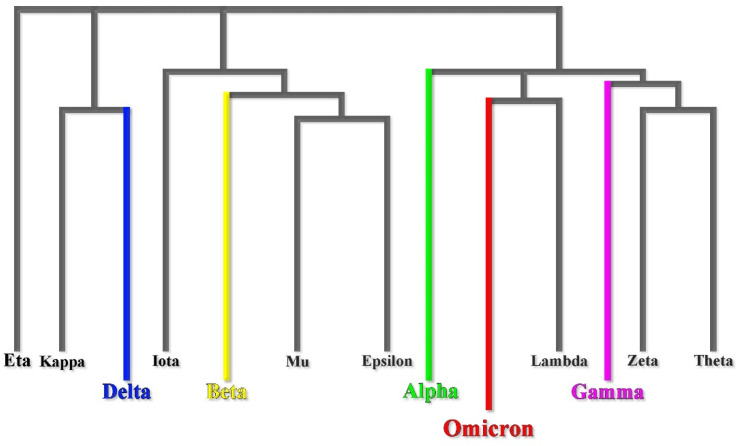
Omicron early divergence in a phylogenetic mutational tree.

**Figure 2 ijms-23-01987-f002:**
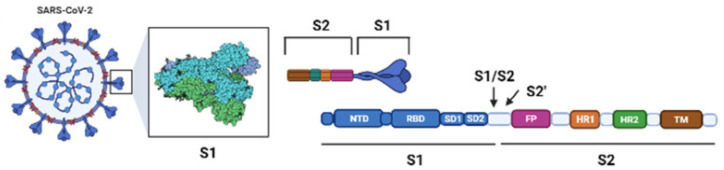
Schematic SARS-CoV-2 and its S protein.

**Figure 3 ijms-23-01987-f003:**
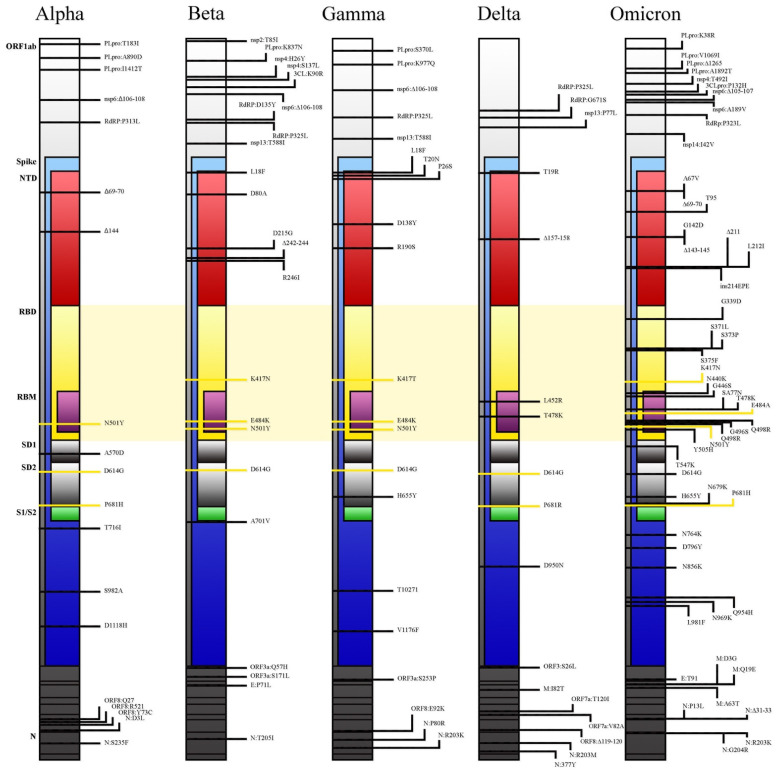
Comparison among mutational profiles of the five already known variants of concern (VOCs).

**Table 1 ijms-23-01987-t001:** Mutational differences among Delta and Omicron variants of concern.

	Mutations	
Variant of Concern	**RBD** (Receptor-Binding Domain)	**NTD** (N-Terminal Domain)
Delta (B.1.617.2)	L452R T478K	T19R T95I G142D ∆E156 ∆F157 R158G
Omicron (B.1.1.529)	G339D S371L S373P S375F K417N N440K G446S S477N T478K E484A Q493R G469S Q498R N501Y Y505H T547K	A67V ∆H69 ∆V70 T95I G142D ∆V143 ∆Y144 ∆Y145 ∆N211 L212I +214EPE

## Data Availability

Not applicable.

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
