# Peer review of "Omicron Genetic and Clinical Peculiarities That May Overturn SARS-CoV-2 Pandemic: A Literature Review"

_ijms, 2022, doi:10.3390/ijms23041987_

Round 1
Reviewer 1 Report
A well-written, thorough and organized manuscript. It presents all the important aspects of the topic that may be of interest to readers. A definitely useful summary in this highly competitive field.
Minor advices:
- A figure presenting the structure of the virus would be useful for non-professional readers.
- Refer to the figures in the text.
- The two paragraphs on page 2 that begin with the sentence “ Recently, whole genome examination and mutational analysis have found ” are hardly understandable.
- If the authors could compare high-affinity antibody binding (e.g., as measured for toxoids) with Spike affinity to ACE2, it would help readers to understand the difficulties of neutralizing the SARS-CoV-2 virus.
- Reference is missing following the sentence: „Furthermore, the Omicron variant shares N501Y, which is believed to increase the binding affinity between the viral spike protein and the angiotensin-converting enzyme 2 (ACE2) receptor.”
- Please indicate more clearly that references 27 and 29 do not apply to the Omicron variant.
- In “Conclusions,” I would avoid the term herd immunity, preferring to use a term like “it reduces the speed of spreading” if the authors agree. (In a condition where protection appears to be short-lived and vaccines do not lead to sterilizing immunity, the meaning of “herd immunity” is very uncertain.)
Reviewer 2 Report
In the review titled ‘Omicron genetic and clinical peculiarities that may overturn SARS-CoV-2 pandemic: a literature review’ the authors have given a detailed account of the current pandemic crisis caused by the Omicron variant. The authors describe the various mutations, transmissibility, immune evasion, epidemiology, clinical manifestations, diagnosis, treatment and future direction. The review is informative, comprehensive and of high quality. There are few points that could be included to make the review article more comprehensive.
- The authors have compared the transmissibility and mutations of the omicron variant with that of delta. It would be useful to compare these characteristics with other VOCs as well and present it in the form of a table.
- Effect of the omicron variant on immunocompromised and patients with co-morbidities as compared to Delta if not other VOCs.
- Effect of second vs third booster vaccine shot on Omicron infectivity and comparison with that of delta.
